# Cost-Effective and Facile Fabrication of a Tattoo Paper-Based SERS Substrate and Its Application in Pesticide Sensing on Fruit Surfaces

**DOI:** 10.3390/nano13030486

**Published:** 2023-01-25

**Authors:** Pratiksha P. Mandrekar, Mingu Kang, Inkyu Park, Bumjoo Kim, Daejong Yang

**Affiliations:** 1Department of Future Convergence Engineering, Kongju National University, Cheonan 31080, Republic of Korea; 2Department of Mechanical Engineering, Korea Advanced Institute of Science and Technology, Daejeon 34141, Republic of Korea; 3Department of Mechanical and Automotive Engineering, Kongju National University, Cheonan 31080, Republic of Korea

**Keywords:** flexible SERS substrates, Raman spectroscopy, silver nanoparticles, thiram, ultraviolet light decomposition

## Abstract

Surface-enhanced Raman spectroscopy (SERS) has been transformed into a useful analytical technique with significant advantages in relation to sensitive and low-concentration chemical analyses. However, SERS substrates are expensive and the analyte sample preparation is complicated; hence, it is only used in limited areas. We have fabricated a tattoo paper-based SERS substrate by using non-complicated inkjet printing. The sensitivity of the SERS substrate was increased by removing the carbon residues via exposure to ultraviolet light without damaging the substrate. Thus, low concentrations of pesticides (up to 1 μM thiram) were measured. The SERS substrate was attached to the curved surface of an apple to demonstrate its advantages, such as the flexibility and easy attachability of tattoo paper, and its feasibility was verified by measuring 1 μM thiram on the apple’s surface. Due to its economic cost, simple usage, and rapid measurement, it will be helpful for the identification of both agricultural adulterants and food adulterants and for water-based pollutant detection. It will also possibly be helpful for medical purposes related to human body surfaces in the future.

## 1. Introduction

Today, as the world is growing at a fast pace in terms of developments and technological advancements, and with a great amount of success, it can be seen that various factors affect living beings on this planet, such as pollution, waste, and other hazardous elements, including pesticides and chemical adulterants, as far as the food industry is concerned. Human health depends on the consumption of various food materials, which are grown naturally and are agricultural products. These food materials are usually laced with various forms of pesticides and insecticides, which are used for crop safety and growth and which often remain as a residue. Although minute consumption of these products does not have any severe or immediate effect on living beings, long-term consumption can cause severe illnesses and harms [1,2,3]. Therefore, it is a requirement to develop a system able to detect the harmful components of food materials and, furthermore, continuously monitor them during culture and production processes.

In line with this trend, in this study, we present a facile and cost-effective way of measuring pesticides on fruit surfaces. Gas chromatography (GC), liquid chromatography (LC), gas chromatography-mass spectrometry (GC/MS), and liquid chromatography-mass spectrometry (LC-MS) are used to analyze the types and amounts of pesticide residues [4,5,6,7,8]. These methods separate an analyte into individual substances via the viscous force or inertial force and then measure the amounts of each substance. Although this method is very precise and highly reliable, it requires very large or complicated components for the separation process. Therefore, these kinds of equipment are large and expensive, and they take a long time to perform the measurements, making them unsuitable for use in the field, such as orchards. As an alternative, the enzyme-linked immunosorbent assay (ELISA) has been proposed, although it is also complicated and time-consuming [9,10,11,12]. For rapid and facile measurements, electrochemical or optical sensors using enzymes, aptamers, antibodies, cells, etc., are being widely developed [13,14,15]. However, these sensors also require pre-processes for the measurements, such as collecting samples and chopping them into extract analytes. Some methods collect analytes from the surface by rubbing or pressing the fruit surface. These methods are also cumbersome and, due to the poor repeatability, the sampling itself reduces the accuracy [16].

Therefore, a method for directly measuring the pesticides on fruit surfaces needs to be developed. Surface-enhanced Raman spectroscopy (SERS) is based on molecular vibration as the signature in the spectra and electromagnetic amplification by conductive nanostructures [17,18,19]. Raman spectroscopy provides specific information concerning the analyte as the fingerprint region of the molecule, thereby providing information on the presence of the molecule [20]. Therefore, unlike conventional methods using enzymes, aptamers, antibodies, or cells, various analytes can be measured without specifying the target in advance. However, since Raman signals are inherently weak, SERS, which can achieve high sensitivity combined with electromagnetic amplification, was developed [17,18,19,21,22]. Conductive NP clusters focus irradiated lasers and generate high plasmonic hotspots that allow SERS to detect even a single molecule [23,24,25]. Since they are based on electromagnetic amplification using conductive nanostructures, SERS devices are usually fabricated using sophisticated micro/nanostructure fabrication processes based on rigid substrates, such as silicon and quartz [20,26,27]. Although extremely high sensitivity is achieved, they are designed to be suitable for measurements in laboratories and have limitations in their applications on work sites. Due to the high sensitivity of SERS, methods for detecting pesticides based on SERS have also been developed [28,29].

Even with SERS, the sample collection processes are a fundamental drawback. Therefore, they have the same problems as chromatography and electrochemical and optical sensing [28,29]. Methods for omitting the analyte preparation processes need to be developed.

Recently, high-performance SERS substrates using randomized high-density nanostructures have been developed, and they do not require sophisticated fabrication methods [30,31,32]. Since this type of device does not need to be limited to rigid substrates, many flexible SERS substrates have been developed, thereby broadening the utilization of SERS [33,34,35]. Moreover, the production cost is also very low.

Various studies have been conducted in order to utilize these flexible SERS substrates for the detection of various components on fruit surfaces, although the majority of these methods involve pasting a substrate onto the surface of the fruit and then peeling it off, or rubbing and then peeling it off [36,37], which may result in low accuracy in terms of the detection of chemicals, as they cannot guarantee the amounts of chemicals transferred to the SERS substrate from the fruit surface. Hence, to overcome these drawbacks and ensure easier attachment, we selected tattoo paper as a substrate, which simply required attachment to the surface of the fruit followed by direct detection on the surface itself. The schematic illustration in Figure 1 shows the application of this work. A piece of tattoo paper-based SERS substrate is attached to an apple in an orchard. Several drops of water used in the attachment process lead to close contact between the silver nanoparticles (AgNPs) and pesticide molecules. The types and concentrations of the pesticides on the fruit surfaces are measured in real time and at work sites using a portable Raman machine.

## 2. Materials and Methods

### 2.1. Chemicals and Reagents

The chemicals and materials used in this work included silver ink (NBSIJ-MU01, Mitsubishi Paper Mills, Sumida, Japan), ethanol (CH_3_CH_2_OH, 99% Daejung Chemicals & Metals Co., Ltd., Siheung, Republic of Korea), benzenethiol (BT, C_6_H_5_SH, ≥99%, Sigma-Aldrich, St. Louis, MO, USA), thiram (C_6_H_12_N_2_S_4_, PESTANAL^®^, analytical standard, Sigma-Aldrich, St. Louis, MO, USA), and tattoo paper (Silhouette America, Inc., Lindon, UT, USA).

### 2.2. Fabrication of Tattoo Paper-Based AgNP SERS Substrates

The AgNP ink was printed onto the tattoo paper using an inkjet printer (Omni-100, Unijet, Seongnam, Republic of Korea). Figure 2 shows the fabrication procedure for the tattoo paper-based AgNP SERS substrates. The protective layer of the tattoo paper was detached, and the carrier layer was directed downward so that the silver nanoparticle ink was printed onto the upper adhesive layer. Patterns of AgNPs with a diameter of 2 mm at a spacing of 10 mm were printed. The AgNP printed substrates were heated at 60 °C on a hotplate to ensure the complete drying of the ink on the tattoo paper. Since AgNP ink contains organic solvents such as ethylene glycol [38], it often results in the formation of carbon residue or carbon byproduct, which tends to remain in the fabricated SERS substrates. In order to remove these carbon-based materials, the substrates are usually heated at over 300 °C, although at such a high temperature, the AgNPs may be agglomerated, and it is intrinsically impossible to apply this process to tattoo paper.

Instead of thermal decomposition, an ultraviolet (UV) light decomposition process was applied to the tattoo paper-based AgNP SERS substrates. A 254 nm wavelength UV light of 300 μW/cm^2^ intensity generated using a 4 W lamp (TN-4LC, Korea Ace Sci., Seoul, Republic of Korea) was irradiated to the SERS substrate for 60 h.

After the carbon decomposition process, the AgNP patterns were diced to a size of 10 mm × 10 mm for convenient use.

### 2.3. Analysis of Tattoo Paper-Based AgNP SERS Substrates

In order to verify the removal of the residual carbon, the SERS spectra were collected randomly at 10 points on the substrate every 5 h throughout the 60 h of the UV light decomposition process. In addition, the amounts of carbon before and after the UV light irradiation were quantitatively measured using energy-dispersive X-ray spectroscopy (EDS). Due to the charging effect of the tattoo paper, glass substrate-based samples were fabricated for the EDS.

The standard solutions of benzenethiol molar concentrations 1 μM and 1 mM were prepared using 99% ethanol as a diluent. The tattoo paper-based AgNP SERS substrates were incubated in the prepared solutions for 24 h and then air-dried. Using a Raman spectrometer (XperRAM C Series, Nanobase Inc., Seoul, Republic of Korea) with a 20× magnitude objective lens and 52.2 mW of a 633 nm laser, the identifications of the benzenethiol (1 μM and 1 mM concentrations) were carried out.

### 2.4. SERS Spectrum Measurement of Thiram

The 1, 10, 100, 200, and 600 μM and 1 mM thiram standard solutions were prepared using 99% ethanol as a diluent. In the same manner as with the benzenethiol, the SERS substrates were incubated in the thiram solutions of six different concentrations, and the SERS spectrum of each sample was measured.

### 2.5. SERS Spectrum Measurement of Thiram Using Real Fruit

Fresh Fuji apples were purchased from a grocery store. They were washed and cleaned thoroughly and repeatedly (4–5 times) using deionized water to ensure that any residual pesticides, if present, were completely removed. Once the apples were dried clean, they were subjected to high-pressured jet air cleaning in order to eliminate any possible foreign residues or dust particles, thereby creating a controlled experimental environment. The apples were then subjected to the solutions of thiram at concentrations of 1, 10, 100, 200, and 600 μM and 1 mM individually. A piece of the tattoo SERS sheet was placed on an apple and a small amount of water was supplied to wet the substrate. The carrier layer was removed, leaving only the tattoo layer on which the AgNPs were printed, and the concentrations of thiram were measured by collecting the SERS signals.

## 3. Results and Discussion

### 3.1. Characteristics of Tattoo Paper-Based AgNP SERS Substrates

The AgNP ink contains ethylene glycol to facilitate printing. The carbons in ethylene glycol have two main Raman peaks at 1320 cm^−1^ (the first-order D mode) and 1580 cm^−1^ (the first-order G mode) [39]. If a large amount of carbon residue remains, it overlaps with the main Raman peak of thiram, which is located at 1382 cm^−1^ [40,41], preventing accurate measurement. In order to remove these carbon components, heating at a high temperature or immersing in an organic solvent are commonly used. However, the tattoo substrate used in this work could not withstand high temperatures or harsh chemical atmospheres. In addition, not only could the substrate be damaged, but also the AgNPs could be aggregated together or separated from the substrate, which would degrade the performance of the SERS substrate. Thus, we used irradiated UV light, which is another well-known method for decomposing carbon compounds [42,43,44]. Moreover, AgNPs used as SERS particles can also act as a catalyst for carbon decomposition [45,46]. The UV lights were irradiated for a sufficiently long time (60 h) to perfectly remove the carbon residue. The SERS signals were measured at 10 random points on the SERS substrate every 5 h. Figure 3a shows the SERS spectra of the substrates before and during 60 h of UV light exposure. The carbon peaks at 1320 cm^−1^ and 1580 cm^−1^ were observed to be decreased dramatically to less than one-tenth, and the background signal was also greatly reduced. As shown in Figure 3b, a small amount of carbon remained after 40 h of irradiation. Even so, in this work, the SERS substrates were irradiated for 60 h to make reliable substrates. A 4 W UV lamp was used and the temperature increase in the substrate during the irradiation was neglectable, meaning that using a high-power UV lamp would reduce the decomposition time. The decrease in the carbon residue was indirectly proved by the intensity of the SERS signal. Thus, the EDS was measured for quantitative verification. The EDS data before and after 60 h of UV light exposure also proved the decrease in the carbon components (Figure 3c).

Figure 4a shows the AgNP patterns on the tattoo substrate. It was printed with a 2 mm diameter and a spacing of 10 mm, and it was diced into 10 mm × 10 mm cells for convenient use. As shown in the scanning electron microscope (SEM) images in Figure 4b–d, the AgNPs were macroscopically uniformly coated on the tattoo paper, and the microscopically rough surfaces of the AgNPs clusters were observed. The size of each particle was measured using an image analysis program (ImageJ, National Institutes of Health, USA). The average diameter was 19.5 nm and the standard deviation was 3.94 nm. The rough surfaces of the AgNP clusters concentrated the electromagnetic field and amplified the Raman signal.

### 3.2. Identification of Benzenethiol and Thiram

In order to evaluate the sensitivity of the tattoo paper-based SERS substrates, the diced substrates were incubated in 1 μM and 1 mM benzenethiol solutions. As shown in Figure 5a, five clear SERS peaks were observed from both the 1 μM and 1 mM benzenethiol solutions. Four peaks came from the benzenethiol and one came from the tattoo paper. These peaks were located at 999 cm^−1^ (in-plain ring-breathing), 1022 cm^−1^ (in-plain C−H bending), 1072 cm^−1^ (C−S stretching), 1410 cm^−1^ (peak from the tattoo paper), and 1574 cm^−1^ (C−C stretching), and they corresponded to the main peaks of benzenethiol [31,47]. These large SERS peaks prove that the tattoo paper-based substrates can be used as a SERS substrate for detecting chemicals.

The various concentrations of thiram solutions were also measured in the same way as the benzenethiol. Thiram can be identified by the SERS peaks located at 560 cm^−1^ (S–S stretching), 933 cm^−1^ (CH_3_N stretching), 1144 cm^−1^ (C–N stretching and CH_3_ rocking), and 1382 cm^−1^ (C–N stretching and symmetric CH_3_ deformation) [40,41,48]. The strongest peak at 1382 cm^−1^ was selected for the quantitative analysis. Figure 5b shows the SERS spectra of the 1, 10, 100, 200, and 600 μM and 1 mM thiram solution. Small but clear peaks were observed at 560, 933 and 1144 cm^−1^, and the largest and most clear peaks were located at 1382 cm^−1^.

According to European Food Safety Authority (EFSA) [49], the acceptable daily intake and acute reference dose of thiram are 0.01 and 0.025 mg/kg/day, respectively. Assuming that a 70 kg person eats an apple with a diameter of 8 cm per day, and that a 1 mm solution film is formed when a pesticide solution is sprayed in an orchard, the acceptable daily intake and acute reference dose of thiram are calculated to be 144.80 and 361.99 μM, respectively. Therefore, 1 μM to 1 mM thiram solutions were used in this experiment.

### 3.3. Identification of Thiram Using Real Fruit

As a feasibility study, we measured the thiram on a fruit’s surface. For reliable measurement, apples purchased from a grocery store were washed clean to remove pesticides and any other impurities on the surface. Various concentrations of thiram solutions were dropped onto the surface of the apples and dried to reproduce pesticide residues. Figure 6a,b show schematic illustrations and photos of the measurement procedures. A piece of tattoo paper-based SERS sheet was placed on the surface of an apple and a few drops of water were dropped. The tattoo paper absorbed the water, and the carrier layer and the tattoo layer were gently separated. The AgNPs on the surface of the tattoo layer came into contact with the surface of the apple smeared with thiram, and the tattoo layer covered it. The water used to attach the tattoo paper dissolved the thiram molecules. This dissolved thiram solution was absorbed into the tattoo layer and increased its contact with the AgNP layer. Since the tattoo layer had very high transparency, the laser could pass through it and reach the AgNPs. In addition, since the tattoo layer covered the SERS layer, the sensing layer could be protected from the external friction that may be applied in an orchard.

Figure 6c shows the SERS spectra of the 1, 10, 100, 200, and 600 μM and 1 mM thiram solution applied to the apples. Four clear peaks at 560, 933, 1144, and 1382 cm^−1^ were observed. As shown in the concentration-intensity curves in Figure 6d, the SERS intensity of the three major peaks increased as the concentration of thiram increased. The tattoo paper-based SERS substrate could detect 1 μM thiram on the surface of an apple by analyzing the clearest peak at 1382 cm^−1^.

## 4. Conclusions

In this study, we have fabricated a tattoo paper-based AgNP SERS substrate. AgNPs were printed onto the surface of the tattoo paper using an inkjet printer. This method of utilizing an inkjet printer is not only helpful in ensuring speedy production, but also enables the possibility of mass production. Moreover, it is fabricated at an ambient temperature and pressure, requires a very small quantity of ink, and inexpensive tattoo paper is utilized. Thus, it is a cost-effective technique. We ensured the sensing performance of the tattoo paper-based SERS substrate by measuring as low as 1 μM benzenethiol and thiram standard solutions. To demonstrate its practicality, this substrate was applied to fruit. Being flexible and adhesive, the tattoo paper was effortlessly attached to an apple’s surface. It succeeded in detecting as low as 1 μM thiram on the apple’s surface. Further studies are needed to develop more sensitive SERS substrates for lower detection limits, to diversify the detectable chemicals, and to elaborate the quantification processes.

As this study presents a novel method of attaching SERS substrates onto curved surfaces, it can be used to analyze materials on the surface of various fruits and vegetables. In addition, since tattoo paper-based SERS substrates are considerably environment-friendly and cost-effective, very easy to handle, and involve simpler sample collection methodology, they can be widely used for onsite pollutant detection in water and other chemical-based food adulterants and additives in future works and applications. Furthermore, as tattoo paper can be easily attached to human and animal skin, it is expected to expand its use for medical purposes in the near future.

## Figures and Tables

**Figure 1 nanomaterials-13-00486-f001:**
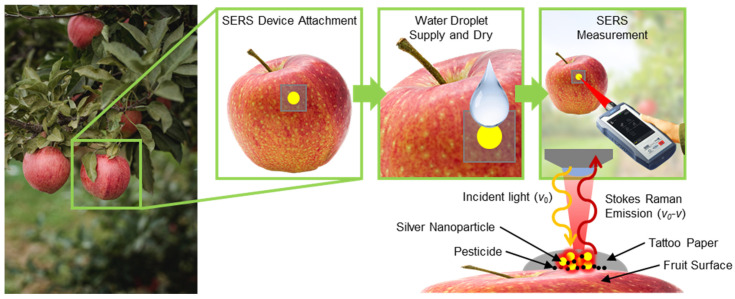
Application of the tattoo paper-based AgNP SERS substrates on a real fruit (apple) sample and identification of the pesticide using a Raman spectrometer.

**Figure 2 nanomaterials-13-00486-f002:**
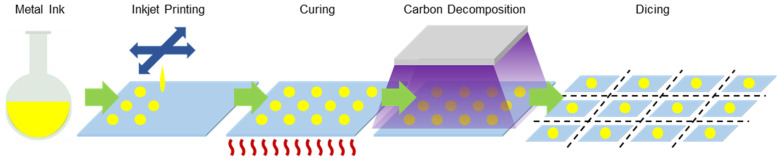
Fabrication steps for the tattoo paper-based AgNP SERS substrates.

**Figure 3 nanomaterials-13-00486-f003:**
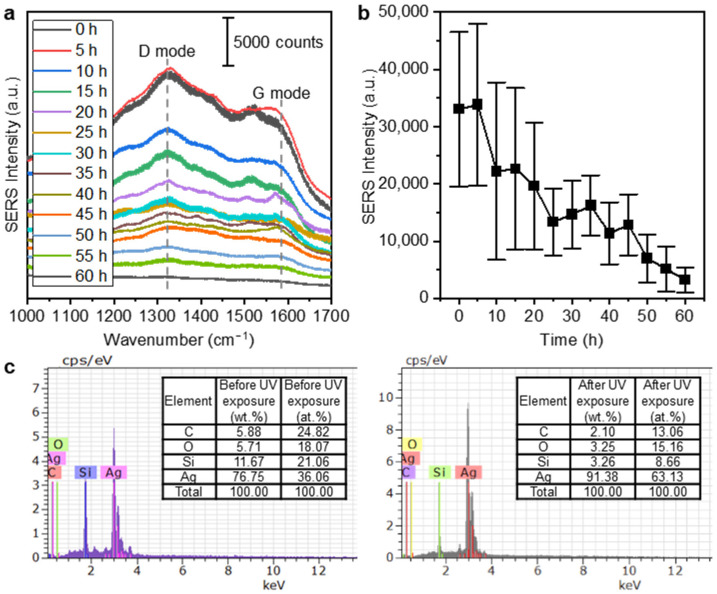
(**a**) SERS intensity spectra of substrates showing gradual carbon decomposition by UV irradiation within the time period from 0 to 60 h; (**b**) reduction in carbon intensity during UV irradiation; and (**c**) EDS data before and after the substrates were subjected to UV irradiation.

**Figure 4 nanomaterials-13-00486-f004:**
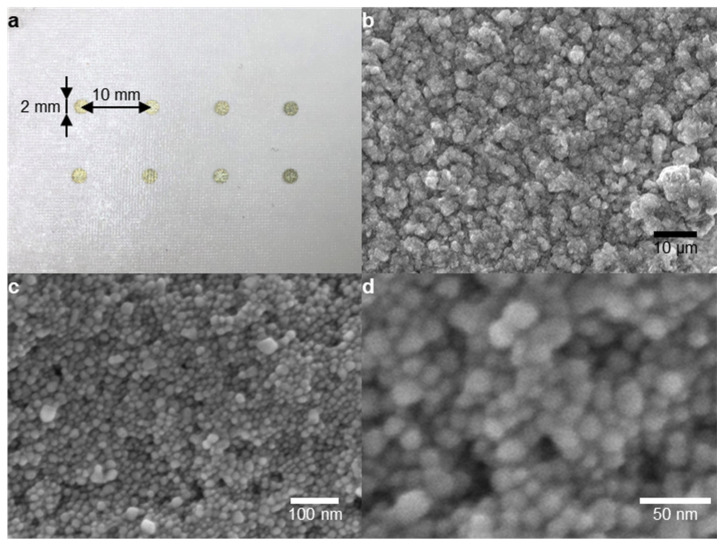
(**a**) Photo of the printed AgNP dot array on tattoo paper; and (**b**–**d**) SEM images of the AgNP clusters printed on the tattoo paper substrate.

**Figure 5 nanomaterials-13-00486-f005:**
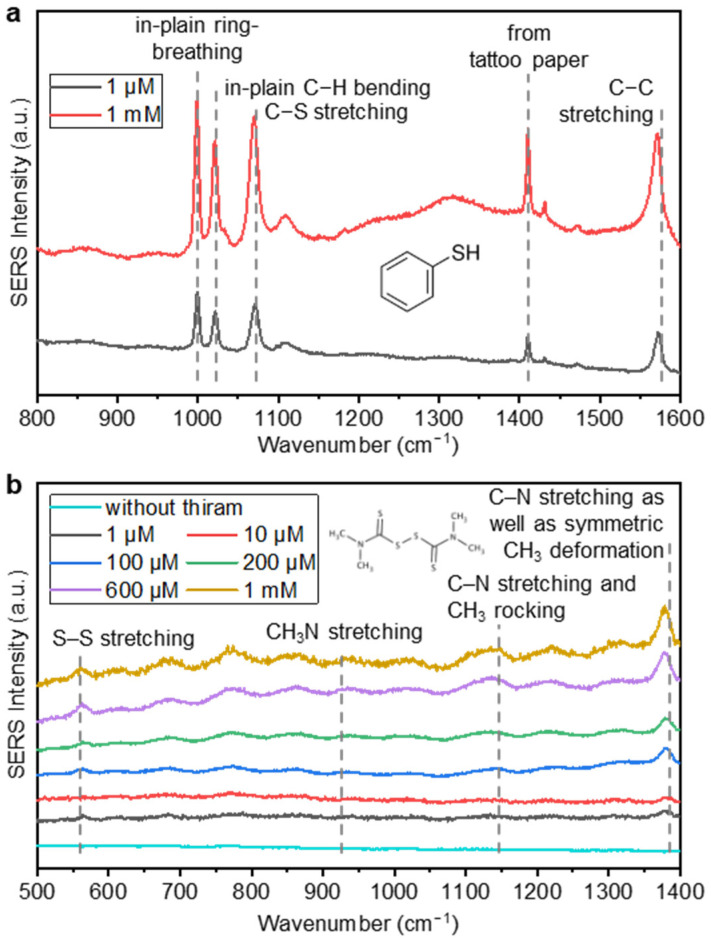
(**a**) SERS intensities at 999, 1022, 1072, 1410, and 1574 cm^−1^, which were measured during the identification of the 1 μM and 1 mM concentrations of benzenethiol solutions; and (**b**) SERS intensity spectrum without thiram and with thiram at concentrations of 1, 10, 100, 200, and 600 μM and 1 mM, exhibiting peak identification at 560, 933, 1144, and 1382 cm^−1^ for all the concentrations.

**Figure 6 nanomaterials-13-00486-f006:**
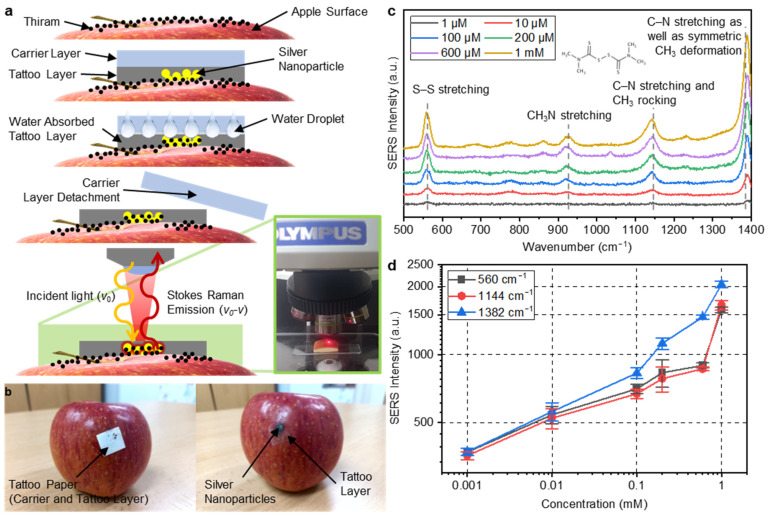
(**a**) Working mechanism of fabricated tattoo paper-based AgNP SERS substrates using a thiram contaminated fruit (apple) sample; (**b**) real apple specimen (thiram contaminated) used for the analysis; (**c**) SERS intensity spectra of the apple specimens showing peak intensities at 560, 933, 1144, and 1382 cm^−1^ for the different concentration of 1, 10, 100, 200, and 600 μM and 1 mM of thiram solutions; and (**d**) intensity curve showing increasing absorption at the 560, 1144, and 1382 cm^−1^ peaks with the higher concentration level of thiram solutions.

## Data Availability

Data are available from the corresponding author upon request.

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
