# Peer review of "Cost-Effective and Facile Fabrication of a Tattoo Paper-Based SERS Substrate and Its Application in Pesticide Sensing on Fruit Surfaces"

_nanomaterials, 2023, doi:10.3390/nano13030486_

Round 1
Reviewer 1 Report
Journal: nanomaterials
Title: Cost Effective and Facile Fabrication of Tattoo Paper-Based SERS Substrates and its Application in Pesticides Sensing on Fruit Surfaces
In this paper, authors design tattoo paper-based SERS substrates to detect thiram on apple surface. AgNPs are used to be the SERS substrates. UV light decomposition process is employed to remove the residual carbon. Overall, the innovation of the manuscript is not clearly articulated and lacks some data to support the application capabilities of the constructed SERS substrate. Therefore, the manuscript is recommended for major revision.
1. The detection of pesticides on fruit surfaces with flexible SERS substrates has been widely reported. The significant improvements and innovations of this manuscript compared to them need to be pointed out, such as materials, detection limits, etc. It is suggested to compare with other methods reported in the literature to illustrate the advantages of this method.
2. The reproducibility of the SERS substrate for pesticide detection is critical for practical application. Please add the corresponding data and calculate RSD to illustrate the reproducibility of the substrate.
3. Authors only used tattoo paper-based SERS substrate to simply detect apples immersed in different concentrations of thiram, however, quantitative data on apple surface pesticides were not examined. The authors need to add this data and verify the accuracy of the SERS analysis using standard analytical methods.
4. The minimum detection limit of tattoo paper-based SERS substrate for thiram on apple surface needs to be investigated. It is also suggested to add the maximum amount of pesticides allowed in fruits to the manuscript.
5. Please add the mapping diagram to further explain the change of carbon composition before and after 60 hours of ultraviolet radiation.
6. Please add the Conclusions section. It must be fully supported by the results reported and should include the major conclusions, the limitations of the work and the future work.
7. Please carefully examine the format of the references, such as the full name or abbreviations of the unified journal.
8. Some detailed explanations need to be added to the manuscript. For example, why water used in the attachment process can close contact between silver nanoparticles and pesticide molecules? What is the mechanism of UV light decomposition process?
Reviewer 2 Report
Please find attached the reviewer comments.

Round 2
Reviewer 1 Report
The authors have taken the reviewers' comments seriously and revised the manuscript accordingly. In my opinion, the revised manuscript could be considered for publication
Reviewer 2 Report
The authors have satisfactorily addressed the reviewer comments, and may be considered for acceptance.